# MULTI-HOP ATTENTION GRAPH NEURAL NETWORKS

## ABSTRACT

Self-attention mechanism in graph neural networks (GNNs) led to state-of-the-art performance on many graph representation learning task. Currently, at every layer, attention is computed between connected pairs of nodes and depends solely on the representation of the two nodes. However, such attention mechanism does not account for nodes that are not directly connected but provide important network context, which could lead to improved predictive performance. Here we propose *Multi-hop Attention Graph Neural Network (MAGNA)*, a principled way to incorporate multi-hop context information into attention computation, enabling long-range interactions at every layer of the GNN. To compute attention between nodes that are not directly connected, MAGNA diffuses the attention scores across the network, which increases the "receptive field" for every layer of the GNN. Unlike previous approaches, MAGNA uses a diffusion prior on attention values, to efficiently account for all paths between the pair of disconnected nodes. This helps MAGNA capture large-scale structural information in every layer, and learn more informative attention. Experimental results on node classification as well as the knowledge graph completion benchmarks show that MAGNA achieves state-of-the-art results: MAGNA achieves up to $5.7\%$ relative error reduction over the previous state-of-the-art on Cora, Citeseer, and Pubmed. MAGNA also obtains the best performance on a large-scale Open Graph Benchmark dataset. On knowledge graph completion MAGNA advances state-of-the-art on WN18RR and FB15k-237 across four different performance metrics.

## 1 INTRODUCTION

The introduction of the self-attention mechanism (Bahdanau et al., 2015; Vaswani et al., 2017), has pushed the state-of-the-art in many domains including graph presentation learning (Radford et al., 2019; Devlin et al., 2019; Liu et al., 2019a; Lan et al., 2019). Graph Attention Network (GAT) (Veličković et al., 2018) and related models (Li et al., 2018; Wang et al., 2019a; Liu et al., 2019b; Oono & Suzuki, 2020) developed attention mechanism for Graph Neural Networks (GNNs), which compute attention scores between nodes connected by an edge, allowing the model to attend to messages of node's direct neighbors according to their attention scores.

However, such attention computation on pairs of nodes connected by edges implies that a node can only attend to its immediate neighbors to compute its (next layer) representation. This implies that receptive field of a single GNN layer is restricted to one-hop network neighborhoods. Although stacking multiple GATs could in principle enlarge the receptive field and learn non-neighboring interactions, such deep GAT architectures suffer from the oversmoothing problem (Wang et al., 2019a; Liu et al., 2019b; Oono & Suzuki, 2020) and do not perform well in practice. Furthermore, edge attention weights in the single GAT layer are based solely on representations of the two nodes at the edge endpoints, and do not depend on their graph neighborhood context. In other words, the one-hop attention mechanism in GATs limits their ability to explore the relationship between the broader graph structure and the attention weights. While previous works (Xu et al., 2018; Klicpera et al., 2019b) have shown advantages in performing multi-hop message-passing in a single layer, these approaches are not graph-attention based. Therefore, incorporating multi-hop neighboring context into the attention computation in graph neural networks had not been explored.

Here we present *Multi-hop Attention Graph Neural Network (MAGNA)*, an effective and efficient multi-hop self-attention mechanism for graph structured data. MAGNA uses a novel graph attention diffusion layer (Figure 1), where we first compute attention weights on edges (represented by solid arrows), and then compute self-attention weights (dotted arrows) between disconnected pairs of nodes through an attention diffusion process using the attention weights on the edges.

Our model has two main advantages: 1) MAGNA captures long-range interactions between nodes that are not directly connected but may be multiple hops away. Thus the model enables effective long-range message passing, from important nodes multiple hops away. 2) The attention computation in MAGNA is context-dependent. The attention value in GATs (Veličković et al., 2018) only depends on node representations of the previous layer, and is zero between disconnected pairs of nodes. In contrast, for any pair of nodes within a chosen multi-hop neighborhood, MAGNA computes attention by aggregating the attention scores over all the possible paths (length $\geq 1$) connecting the two nodes.

Theoretically we demonstrate that MAGNA places a Personalized Page Rank (PPR) prior on the attention values. We further apply spectral graph analysis to show that MAGNA has the capability of emphasizing on large-scale graph structure and lowering high-frequency noise in graphs. Specifically, MAGNA enlarges the lower Laplacian eigen-values, which correspond to the large-scale structure in the graph, and suppresses the higher Laplacian eigen-values which correspond to more noisy and fine-grained information in the graph.

We experiment on standard datasets for semi-supervised node classification as well as knowledge graph completion. Experiments show that MAGNA achieves state-of-the-art results: MAGNA achieves up to $5.7\%$ relative error reduction over previous state-of-the-art on Cora, Citeseer, and Pubmed. MAGNA also obtains better performance on a large-scale Open Graph Benchmark dataset. On knowledge graph completion, MAGNA advances state-of-the-art on WN18RR and FB15k-237 across four metrics, with the largest gain of 7.1% in the metric of Hit at 1.

Furthermore, we show that MAGNA with just 3 layers and 6 hop wide attention per layer significantly out-performs GAT with 18 layers, even though both architectures have the same receptive field. Moreover, our ablation study reveals the synergistic effect of the essential components of MAGNA, including layer normalization and multi-hop diffused attention. We further observe that compared to GAT, the attention values learned by MAGNA have higher diversity, indicating the ability to better pay attention to important nodes.

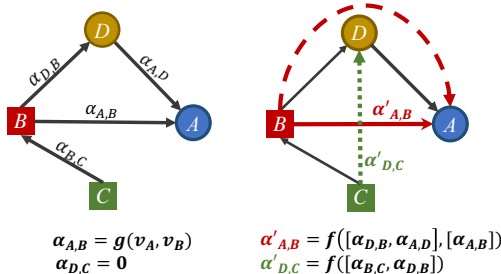

$$\alpha_{A,B} = g(v_A, v_B) \qquad \alpha'_{A,B} = f([\alpha_{D,B}, \alpha_{A,D}], [\alpha_{A,B}])$$
$$\alpha_{D,C} = 0 \qquad \alpha'_{D,C} = f([\alpha_{B,C}, \alpha_{D,B}])$$

Figure 1: **Multi-hop attention diffusion.** Consider making a prediction at nodes $A$ and $D$. **Left**: A single GAT layer only computes attention scores $\boldsymbol{\alpha}$ between directly connected pairs of nodes (i.e., edges) and thus $\boldsymbol{\alpha}_{D,C} = 0$. Furthermore, the attention $\boldsymbol{\alpha}_{A,B}$ between $A$ and $B$ only depends on their node representations. **Right**: A single MAGNA layer is able to: (1) capture the information of two-hop neighbor $C$ to $D$ via the diffused multi-hop attention $\boldsymbol{\alpha}'_{D,C}$; And, (2) enhance graph structure learning by considering all paths between nodes via diffused attention based on powers of graph adjacency matrix.

## 2 Multi-hop Attention Graph Neural Network (MAGNA)

We first discuss the background and then explain Multi-hop Attention Graph Neural Network's novel multi-hop attention diffusion module and its overall model architecture.

### 2.1 Preliminaries

Let $\mathcal{G} = (\mathcal{V}, \mathcal{E})$ be a given graph, where $\mathcal{V}$ is the set of $N_n$ nodes, $\mathcal{E} \subseteq \mathcal{V} \times \mathcal{V}$ is the set of $N_e$ edges connecting $M$ pairs of nodes in $\mathcal{V}$. Each node $v \in \mathcal{V}$ and each edge $e \in \mathcal{E}$ are associated with their type mapping functions: $\phi : \mathcal{V} \to \mathcal{T}$ and $\psi : \mathcal{E} \to \mathcal{R}$. Here $\mathcal{T}$ and $\mathcal{R}$ denote the sets of node types (labels) and edge/relation types. Our framework supports learning on heterogeneous graphs with multiple elements in $\mathcal{R}$.

A general Graph Neural Network (GNN) approach learns an embedding that maps nodes and/or edge types into a continuous vector space. Let $\boldsymbol{X} \in \mathbb{R}^{N_n \times d_n}$ and $\boldsymbol{R} \in \mathbb{R}^{N_r \times d_r}$ be the node embedding and edge/relation type embedding, where $N_n = |\mathcal{V}|$, $N_r = |\mathcal{R}|$, $d_n$ and $d_r$ represent the embedding dimension of node and edge/relation types, each row $\boldsymbol{x}_i = \boldsymbol{X}[i :]$ represents the embedding of node $v_i$ ($1 \leq i \leq N_n$), and $\boldsymbol{r}_j = \boldsymbol{R}[j :]$ represents the embedding of relation $r_j$ ($1 \leq j \leq N_r$).

MAGNA builds on GNNs, while bringing together the benefits of Graph Attention and Diffusion techniques. The core of MAGNA is *Multi-hop Attention Diffusion*, a principled way to learn attention between any pair of nodes in a scalable way, taking into

account the graph structure and enabling multi-hop context-dependent attention directly.

The key challenge here is how to allow for flexible but scalable context-dependent multi-hop attention, where any node can influence embedding of any other node in a single GNN layer (even if they are far away in the underlying network). Simply learning attention scores over all node pairs is infeasible and would lead to overfitting and poor generalization.

## 2.2 MULTI-HOP ATTENTION DIFFUSION

We first introduce attention diffusion to compute the multi-hop attention directly, which operates on the MAGNA's attention scores at each layer. The input to the attention diffusion operator is a set of triples $(v_i, r_k, v_j)$, where $v_i, v_j$ are nodes and $r_k$ is the edge type. MAGNA first computes the attention scores on all edges. The attention diffusion module then computes the attention values between pairs of nodes that are not directly connected by an edge, based on the edge attention scores, via a diffusion process. The attention diffusion module can then be used as a component in MAGNA architecture, which we will further elaborate in Section 2.3.

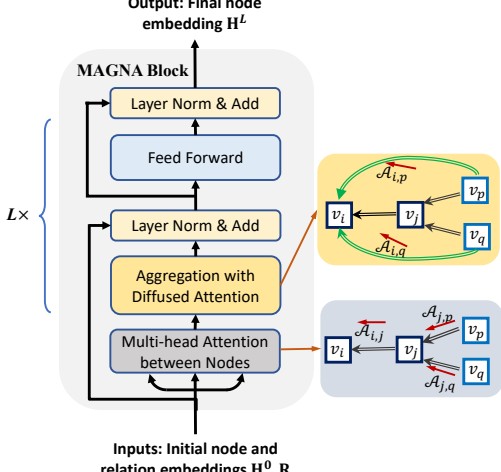

Figure 2: **MAGNA Architecture**. Each MAGNA block consists of attention computation, attention diffusion, layer normalization, feed forward layers, and 2 residual connections for each block. MAGNA blocks can be stacked to constitute a deep model. As illustrated on the right, context-dependent attention is achieved via the attention diffusion process. Here $v_i, v_j, v_p, v_q \in \mathcal{V}$ are nodes in the graph.

**Edge Attention Computation**. At each layer $l$, a vector message is computed for each triple $(v_i, r_k, v_j)$. To compute the representation of $v_j$ at layer $l + 1$, all messages from triples incident to $v_j$ are aggregated into a single message, which is then used to update $v_j^{l+1}$.

In the first stage, the attention score $s$ of an edge $(v_i, r_k, v_j)$ is computed by the following:

$$s_{i,k,j}^{(l)} = \text{LeakyReLU}(\boldsymbol{v}_a^{(l)} \tanh(\boldsymbol{W}_h^{(l)}\boldsymbol{h}_i^{(l)} \| \boldsymbol{W}_t^{(l)}\boldsymbol{h}_j^{(l)} \| \boldsymbol{W}_r^{(l)}\boldsymbol{r}_k)) \quad (1)$$

where $\boldsymbol{W}_h^{(l)}, \boldsymbol{W}_t^{(l)} \in \mathbb{R}^{d^{(l)} \times d^{(l)}}$, $\boldsymbol{W}_r^{(l)} \in \mathbb{R}^{d^{(l)} \times d_r}$ and $\boldsymbol{v}_a^{(l)} \in \mathbb{R}^{1 \times 3d^{(l)}}$ are the trainable weights shared by $l$-th layer. $\boldsymbol{h}_i^{(l)} \in \mathbb{R}^{d^{(l)}}$ represents the embedding of node $i$ at $l$-th layer, and $\boldsymbol{h}_i^{(0)} = \boldsymbol{x}_i$. $\boldsymbol{r}_k$ is the trainable relation embedding of the $k$-th relation type $r_k$ ($1 \leq k \leq N_r$), and $a\|b$ denotes concatenation of embedding vectors $a$ and $b$. For graphs with no relation type, we treat as a degenerate categorical distribution with 1 category [1].

Applying Eq. 1 on each edge of the graph $\mathcal{G}$, we obtain an attention score matrix $\boldsymbol{S}^{(l)}$:

$$\boldsymbol{S}_{i,j}^{(l)} = \begin{cases} s_{i,k,j}^{(l)}, & \text{if } (v_i, r_k, v_j) \text{ appears in } \mathcal{G} \\ -\infty, & \text{otherwise} \end{cases} \quad (2)$$

Subsequently we obtain the attention matrix $\boldsymbol{A}^{(l)}$ by performing row-wised softmax over the score matrix $\boldsymbol{S}^{(l)}$: $\boldsymbol{A}^{(l)} = \text{softmax}(\boldsymbol{S}^{(l)})$. $\boldsymbol{A}_{ij}^{(l)}$ denotes the attention value at layer $l$ when aggregating message from node $j$ to node $i$.

**Attention Diffusion for Multi-hop Neighbors**. In the second stage, we further enable attention between nodes that are not directly connected in the network. We achieve this via the following attention diffusion procedure. The procedure computes the attention scores of multi-hop neighbors via graph diffusion based on the powers of the 1-hop attention matrix $\boldsymbol{A}$:

$$\mathcal{A} = \sum_{i=0}^{\infty} \theta_i \boldsymbol{A}^i \text{ where } \sum_{i=0}^{\infty} \theta_i = 1 \text{ and } \theta_i > 0 \quad (3)$$

where $\theta_i$ is the attention decay factor and $\theta_i > \theta_{i+1}$. The powers of attention matrix, $A^i$, give us the number of relation paths from node $h$ to node $t$ of length up to $i$, increasing the receptive field of the attention (Figure 1). Importantly, the mechanism allows the attention between two nodes to not only depend on their previous layer representations, but also taking into account of the paths between the

---

[1]In this case, we can view that there is only one "pseudo" relation type (category), i.e., $N_r = 1$

nodes, effectively creating attention shortcuts between nodes that are not directly connected (Figure 1). Attention through each path is also weighted differently, depending on $\theta$ and the path length.

In our implementation we utilize the geometric distribution $\theta_i = \alpha(1-\alpha)^i$, where $\alpha \in (0,1]$. The choice is based on the inductive bias that nodes further away should be weighted less in message aggregation, and nodes with different relation path lengths to the target node are sequentially weighted in an independent manner. In addition, notice that if we define $\theta_0 = \alpha \in (0,1]$, $\boldsymbol{A}^0 = \boldsymbol{I}$, then Eq. 3 gives the Personalized Page Rank (PPR) procedure on the graph with the attention matrix $\boldsymbol{A}$ and teleport probability $\alpha$. Hence the diffused attention weights, $\mathcal{A}_{ij}$, can be seen as the influence of node $j$ to node $i$. We further elaborate the significance of this observation in Section 4.3.

We can also view $\mathcal{A}_{ij}$ as the attention value of node $j$ to $i$ since $\sum_{j=1}^{N_n} \mathcal{A}_{ij} = 1$.[2] We then define the *graph attention diffusion* based feature aggregation as

$$\text{AttDiffusion}(\mathcal{G}, \boldsymbol{H}^{(l)}, \Theta) = \mathcal{A}\boldsymbol{H}^{(l)}, \tag{4}$$

where $\Theta$ is the set of parameters for computing attention. Thanks to the diffusion process defined in Eq. 3, MAGNA uses the same number of parameters as if we were only computing attention between nodes connected via edges. This ensures runtime efficiency as well as good model generalization.

**Approximate Computation for Attention Diffusion**. For large graphs computing the exact attention diffusion matrix $\mathcal{A}$ using Eq. 3 may be prohibitively expensive, due to computing the powers of the attention matrix (Klicpera et al., 2019a). To resolve this bottleneck, we proceed as follows: Let $\boldsymbol{H}^{(l)}$ be the input entity embedding of the $l$-th layer ($\boldsymbol{H}^{(0)} = \boldsymbol{X}$) and $\theta_i = \alpha(1-\alpha)^i$. Since MAGNA only requires aggregation via $\mathcal{A}\boldsymbol{H}^{(l)}$, we can approximate $\mathcal{A}\boldsymbol{H}^{(l)}$ by defining a sequence $Z^{(K)}$ which converges to the true value of $\mathcal{A}\boldsymbol{H}^{(l)}$ (Proposition 1) as $K \to \infty$:

$$\boldsymbol{Z}^{(0)} = \boldsymbol{H}^{(l)}, \quad \boldsymbol{Z}^{(k+1)} = (1-\alpha)\boldsymbol{A}\boldsymbol{Z}^{(k)} + \alpha\boldsymbol{Z}^{(0)}, \text{ where } 0 \leq k < K \tag{5}$$

**Proposition 1.** $\lim_{K\to\infty} Z^{(K)} = \mathcal{A}\boldsymbol{H}^{(l)}$

In the Appendix we give the proof which relies on the expansion of Eq. 5.

Using the above approximation, the complexity of attention computation with diffusion is still $O(|E|)$, with a constant factor corresponding to the number of hops $K$. In practice, we find that choosing the values of $K$ such that $3 \leq K \leq 10$ results in good model performance. Many real-world graphs exhibit small-world property, in which case even a smaller value of $K$ is sufficient. For graphs with larger diameter, we choose larger $K$, and lower the value of $\alpha$.

### 2.3 DIRECT MULTI-HOP ATTENTION BASED GNN ARCHITECTURE

Figure 2 provides an architecture overview of the MAGNA Block that can be stacked multiple times.

**Multi-head Graph Attention Diffusion Layer**. Multi-head attention (Vaswani et al., 2017; Veličković et al., 2018) is used to allow the model to jointly attend to information from different representation sub-spaces at different viewpoints. In Eq. 6, the attention diffusion for each head $i$ is computed separately with Eq. 4, and aggregated:

$$\hat{\boldsymbol{H}}^{(l)} = \text{MultiHead}(\mathcal{G}, \tilde{\boldsymbol{H}}^{(l)}) = \left(\Big\|_{i=1}^{M} \text{head}_i\right) \boldsymbol{W}_o, \quad \text{where}$$

$$\text{head}_i = \text{AttDiffusion}(\mathcal{G}, \tilde{\boldsymbol{H}}^{(l)}, \Theta_i), \tilde{\boldsymbol{H}}^{(l)} = \text{LayerNorm}(\boldsymbol{H}^{(l)}),$$

$$\tag{6}$$

where $\|$ denotes concatenation and $\Theta_i$ are the parameters in Eq. 1 for the $i$-th head ($1 \leq i \leq M$), and $\boldsymbol{W}_o$ represents a parameter matrix. Since we calculate the attention diffusion in a recursive way in Eq. 5, we add layer normalization which helpful to stabilize the recurrent computation procedure (Ba et al., 2016).

**Deep Aggregation**. Moreover our MAGNA block contains a fully connected feed-forward sub-layer, which consists of a two-layer feed-forward network. We also add the layer normalization and residual connection in both sub-layers, allowing for a more expressive aggregation step for each block (Xiong et al., 2020):

$$\hat{\boldsymbol{H}}^{(l+1)} = \hat{\boldsymbol{H}}^{(l)} + \boldsymbol{H}^{(l)}$$

$$\boldsymbol{H}^{(l+1)} = \boldsymbol{W}_2^{(l)}\text{ReLU}\left(\boldsymbol{W}_1^{(l)}\text{LayerNorm}(\hat{\boldsymbol{H}}^{(l+1)})\right) + \hat{\boldsymbol{H}}^{(l+1)} \tag{7}$$

**MAGNA generalizes GAT**. MAGNA extends GAT via the diffusion process. The feature aggregation in GAT is $\boldsymbol{H}^{(l+1)} = \sigma(\boldsymbol{A}\boldsymbol{H}^{(l)}\boldsymbol{W}^{(l)})$, where $\sigma$ represents the activation function. We can

---

[2]Obtained by the attention definition $\boldsymbol{A}^{(l)} = \text{softmax}(\boldsymbol{S}^{(l)})$ and Eq. 3.

divide GAT layer into two components as follows:

$$H^{(l+1)} = \underbrace{\sigma}_{(2)}(\underbrace{A H^{(2)} W^{(l)}}_{(1)}).$$
(8)

In component (1), MAGNA removes the restriction of attending to direct neighbors, without requiring additional parameters as $\mathcal{A}$ is induced from $A$. For component (2) MAGNA uses layer normalization and deep aggregation which achieves significant gains according to ablation studies in Table 1. Compared to the "shallow" activation function *elu* in GAT, we can view deep aggregation (i.e., two-layer MLP) as a learnable deep activation function as two layer MLP can approximate many different functions (Pinkus, 1999).

## 3 ANALYSIS OF GRAPH ATTENTION DIFFUSION

In this section, we investigate the benefits of MAGNA from the viewpoint of discrete signal processing on graphs (Sandryhaila & Moura, 2013). Our first result demonstrates that MAGNA can better capture large-scale structural information. Our second result explores the relation between MAGNA and Personalized PageRank (PPR).

### 3.1 SPECTRAL PROPERTIES OF GRAPH ATTENTION DIFFUSION

We view the attention matrix $A$ of GAT, and $\mathcal{A}$ of MAGNA as weighted adjacency matrices, and apply Graph Fourier transform and spectral analysis (details in Appendix) to show the effect of MAGNA as a graph low-pass filter, being able to more effectively capture large-scale structure in graphs. By Eq. 3, the sum of each row of either $\mathcal{A}$ or $A$ is 1. Hence the normalized graph Laplacians are $\hat{L}_{sym} = I - \mathcal{A}$ and $L_{sym} = I - A$ for $\mathcal{A}$ and $A$ respectively. We can get the following proposition:

**Proposition 2.** *Let $\hat{\lambda}_i^g$ and $\lambda_i^g$ be the $i$-th eigeinvalues of $\hat{L}_{sym}$ and $L_{sym}$.*

$$\frac{\hat{\lambda}_i^g}{\lambda_i^g} = \frac{1 - \frac{\alpha}{1-(1-\alpha)(1-\lambda_i^g)}}{\lambda_i^g} = \frac{1}{\frac{\alpha}{1-\alpha} + \lambda_i^g}.$$
(9)

Refer to Appendix for the proof. We additionally have $\lambda_i^g \in [0,2]$ (proved by (Ng et al., 2002)). Eq. 9 shows that when $\lambda_i^g$ is small such that $\frac{\alpha}{1-\alpha} + \lambda_i^g < 1$, then $\hat{\lambda}_i^g > \lambda_i^g$, whereas for large $\lambda_i^g$, $\hat{\lambda}_i^g < \lambda_i^g$. This relation indicates that the use of $\mathcal{A}$ increases smaller eigenvalues and decreases larger eigenvalues[3]. See Section 4.3 for its empirical evidence. The low-pass effect increases with smaller $\alpha$.

The eigenvalues of the low-frequency signals describe the large-scale structure in the graph (Ng et al., 2002) and have been shown to be crucial in graph tasks (Klicpera et al., 2019b). As $\lambda_i^g \in [0,2]$ (Ng et al., 2002) and $\frac{\alpha}{1-\alpha} > 0$, the reciprocal format in Eq. 9 will amplify the ratio of lower eigenvalues to the sum of all eigenvalues. In contrast, high eigenvalues corresponding to noise are suppressed.

### 3.2 PERSONALIZED PAGERANK MEETS GRAPH ATTENTION DIFFUSION

We can also view the attention matrix $A$ as a random walk matrix on graph $\mathcal{G}$ since $\sum_{j=1}^{N_n} A_{i,j} = 1$ and $A_{i,j} > 0$. If we perform Personalized PageRank (PPR).with parameter $\alpha \in (0,1]$ on $\mathcal{G}$ with transition matrix $A$, the fully Personalized PageRank (Lofgren, 2015) is defined as:

$$A_{ppr} = \alpha(I - (1-\alpha)A)^{-1}$$
(10)

Using the power series expansion for the matrix inverse, we obtain

$$A_{ppr} = \alpha \sum_{i=0}^{\infty}(1-\alpha)^i A^i = \sum_{i=0}^{\infty}\alpha(1-\alpha)^i A^i$$
(11)

Comparing to the diffusion Equation 3 with $\theta_i = \alpha(1-\alpha)^i$, we have the following proposition.

**Proposition 3.** *Graph attention diffusion defines a personalized page rank with parameter $\alpha \in (0,1]$ on $\mathcal{G}$ with transition matrix $A$, i.e., $\mathcal{A} = A_{ppr}$.*

The parameter $\alpha$ in MAGNA is equivalent to the teleport probability of PPR. PPR provides a good relevance score between nodes in a weighted graph (the weights from the attention matrix $A$). In

---

[3]The eigenvalues of $\mathcal{A}$ and $A$ correspond to the same eigenvectors, as shown in Proposition 2 in Appendix.

Table 1: Node classification accuracy on Cora, Citeseer, Pubmed. MAGNA achieves state-of-the-art.

| | Models | Cora | Citeseer | Pubmed |
|---|---|---|---|---|
| Baselines | GCN (Kipf & Welling, 2016) | 81.5 | 70.3 | 79.0 |
| | Chebyshev (Defferrard et al., 2016) | 81.2 | 69.8 | 74.4 |
| | DualGCN (Zhuang & Ma, 2018) | 83.5 | 72.6 | 80.0 |
| | JKNet (Xu et al., 2018)* | 81.1 | 69.8 | 78.1 |
| | LGCN (Gao et al., 2018) | $83.3 \pm 0.5$ | $73.0 \pm 0.6$ | $79.5 \pm 0.2$ |
| | Diffusion-GCN (Klicpera et al., 2019b) | $83.6 \pm 0.2$ | $73.4 \pm 0.3$ | $79.6 \pm 0.4$ |
| | APPNP (Klicpera et al., 2019a) | $84.3 \pm 0.2$ | $71.1 \pm 0.4$ | $79.7 \pm 0.3$ |
| | g-U-Nets (Gao & Ji, 2019) | $84.4 \pm 0.6$ | $73.2 \pm 0.5$ | $79.6 \pm 0.2$ |
| | GAT (Veličković et al., 2018) | $83.0 \pm 0.7$ | $72.5 \pm 0.7$ | $79.0 \pm 0.3$ |
| Abl. | No LayerNorm | $83.8 \pm 0.6$ | $71.1 \pm 0.5$ | $79.8 \pm 0.2$ |
| | No Diffusion | $83.0 \pm 0.4$ | $71.6 \pm 0.4$ | $79.3 \pm 0.3$ |
| | No Feed-Forward◇ | $84.9 \pm 0.4$ | $72.2 \pm 0.3$ | $80.9 \pm 0.3$ |
| | No (LayerNorm + Feed-Forward) | $84.3 \pm 0.6$ | $72.6 \pm 0.4$ | $79.6 \pm 0.4$ |
| | **MAGNA** | $\mathbf{85.4} \pm 0.6$ | $\mathbf{73.7} \pm 0.5$ | $\mathbf{81.4} \pm 0.2$ |

* : based on the implementation in https://github.com/DropEdge/DropEdge;
◇ : replace the feed forward layer with *elu* used in GAT.

summary, MAGNA places a PPR prior over node pairwise attention scores: the diffused attention between node $i$ and $j$ depends on the attention scores on the edges of all paths between $i$ and $j$.

## 4 EXPERIMENTS

We evaluate MAGNA on two classical tasks[4]. (1) On node classification we achieve an average of 5.7% relative error reduction; (2) On knowledge graph completion we achieve 7.1% relative improvement in the Hit at 1 metric.[5] We compare with numbers reported by baseline papers when available.

### 4.1 TASK 1: NODE CLASSIFICATION

**Datasets**. We employ four benchmark datasets for node classification: (1) standard citation network benchmarks Cora, Citeseer and Pubmed (Sen et al., 2008; Kipf & Welling, 2016); and (2) a benchmark dataset ogbn-arxiv on 170k nodes and 1.2m edges from the Open Graph Benchmark (Weihua Hu, 2020). We follow the standard data splits for all datasets. Further information about these datasets is summarized in the Appendix.

**Baselines**. We compare against a comprehensive suite of state-of-the-art GNN methods including: GCNs (Kipf & Welling, 2016), Chebyshev filter based GCNs (Defferrard et al., 2016), DualGCN (Zhuang & Ma, 2018), JKNet (Xu et al., 2018), LGCN (Gao et al., 2018), Diffusion-GCN (Klicpera et al., 2019b), APPNP (Klicpera et al., 2019a), Graph U-Nets (g-U-Nets) (Gao & Ji, 2019), and GAT (Veličković et al., 2018).

**Experimental Setup**. For datasets Cora, Citeseer and Pubmed, we use 6 MAGNA blocks with hidden dimension 512 and 8 attention heads. For the large-scale ogbn-arxiv dataset, we use 2 MAGNA blocks with hidden dimension 128 and 8 attention heads. Refer to Appendix for detailed description of all hyper-parameters and evaluation settings.

Table 2: Node classification accuracy on the OGB Arxiv dataset.

| Data | GCN (Kipf & Welling, 2016) | GraphSAGE (Hamilton et al., 2017) | Node2vec (Grover & Leskovec, 2016) | JKNet (Xu et al., 2018) | GaAN (Zhang et al., 2018) | MLP | **MAGNA** |
|---|---|---|---|---|---|---|---|
| ogbn-arxiv | $71.74 \pm 0.29$ | $71.49 \pm 0.27$ | $70.07 \pm 0.13$ | $72.19 \pm 0.21$ | $71.97 \pm 0.24$ | $55.50 \pm 0.23$ | $\mathbf{72.76} \pm 0.14$ |

**Results**. We report node classification accuracies on the benchmarks. Results are summarized in Tables 1 and 2. MAGNA improves over all methods and achieves the new state-of-the-art on all datasets.

---

[4] All datasets used are public, and the code will be released at the time of publication.

[5] Please see the definitions of these two tasks in Appendix.

Table 3: KG Completion on WN18RR and FB15k-237. MAGNA achieves state of the art.

| Models | WN18RR | | | | | FB15k-237 | | | | |
|---|---|---|---|---|---|---|---|---|---|---|
| | MR | MRR | H@1 | H@3 | H@10 | MR | MRR | H@1 | H@3 | H@10 |
| TransE (Bordes et al., 2013) | 3384 | .226 | - | - | .501 | 357 | .294 | - | - | .465 |
| RotatE (Sun et al., 2019) | 3340 | .476 | .428 | .492 | .571 | 177 | .338 | .241 | .375 | .533 |
| OTE (Tang et al., 2020) | - | .491 | .442 | .511 | .583 | - | .361 | .267 | .396 | .550 |
| ROTH (Chami et al., 2020) | - | .496 | .449 | .514 | .586 | - | .344 | .246 | .380 | .535 |
| ComplEx (Trouillon et al., 2016) | 5261 | .44 | .41 | .46 | .51 | 339 | .247 | .158 | .275 | .428 |
| QuatE (Zhang et al., 2019) | 2314 | .488 | .438 | .508 | .582 | - | .366 | .271 | .401 | .556 |
| CoKE (Wang et al., 2019b) | - | .475 | .437 | .490 | .552 | - | .361 | .269 | .398 | .547 |
| ConvE (Dettmers et al., 2018) | 4187 | .43 | .40 | .44 | .52 | 244 | .325 | .237 | .356 | .501 |
| DistMult (Yang et al., 2015) | 5110 | .43 | .39 | .44 | .49 | 254 | .241 | .155 | .263 | .419 |
| TuckER (Balazevic et al., 2019) | - | .470 | .443 | .482 | .526 | - | .358 | .266 | .392 | .544 |
| R-GCN (Schlichtkrull et al., 2018) | - | - | - | - | - | - | .249 | .151 | .264 | .417 |
| SACN (Shang et al., 2019) | - | .47 | .43 | .48 | .54 | - | .35 | .26 | .39 | .54 |
| A2N (Bansal et al., 2019) | - | .45 | .42 | .46 | .51 | - | .317 | .232 | .348 | .486 |
| **MAGNA + DistMult** | 2545 | .502 | .459 | .519 | .589 | 138 | .369 | .275 | .409 | .563 |

**Ablation study**. We report (Table 1) the model performance after removing each component of MAGNA (layer normalization, attention diffusion and deep aggregation feed forward layers) from every layer of MAGNA. Note that the model is equivalent to GAT without these three components. We observe that both diffusion and layer normalization play a crucial role in improving the node classification performance for all datasets. While layer normalization alone does not benefit GNNs, its use in conjunction with the attention diffusion module significantly boosts MAGNA's performance. Since MAGNA computes many attention values, layer normalization is crucial in ensuring training stability (Ba et al., 2016). Meanwhile, we also remove both layer normalization and deep aggregation feed forward layer, and only keep the attention diffusion layer (see the next-to-last row of Table 1). Comparing to GAT, attention diffusion allows multi-hop attention in each layer still benefits the performance of node classification.

## 4.2 TASK 2: KNOWLEDGE GRAPH COMPLETION

**Datasets**. We evaluate MAGNA on standard benchmark knowledge graphs: WN18RR (Dettmers et al., 2018) and FB15K-237 (Toutanova & Chen, 2015). See the statistics of these KGs in Appendix.

**Baselines**. We compare MAGNA with state-of-the-art baselines, including (1) translational distance based models: TransE (Bordes et al., 2013) and its latest extension RotatE (Sun et al., 2019), OTE (Tang et al., 2020) and ROTH (Chami et al., 2020); (2) semantic matching based models: ComplEx (Trouillon et al., 2016), QuatE (Zhang et al., 2019), CoKE (Wang et al., 2019b), ConvE (Dettmers et al., 2018), DistMult (Yang et al., 2015) and TuckER (Balazevic et al., 2019); (3) GNN-based models: R-GCN (Schlichtkrull et al., 2018), SACN (Shang et al., 2019) and A2N (Bansal et al., 2019).

**Training procedure**. We use the standard training procedure used in previous KG embedding models (Balazevic et al., 2019; Dettmers et al., 2018) (Appendix for details). We follow an encoder-decoder framework: The encoder applies the proposed MAGNA model to compute the entity embeddings. The decoder then makes link prediction given the embeddings, and existing decoders in prior models can be applied. To show the power of MAGNA, we employ the DistMult decoder (Yang et al., 2015), a simple decoder without extra parameters.

**Evaluation**. We use the standard split for the benchmarks, and the standard testing procedure of predicting tail (head) entity given the head (tail) entity and relation type. We exactly follow the evaluation used by all previous works, namely the Mean Reciprocal Rank (MRR), Mean Rank (MR), and hit rate at $K$ (H@K). See Appendix for a detailed description of this standard setup.

**Results**. MAGNA achieves new state-of-the-art in knowledge graph completion on all four metrics (Table 3). MAGNA compares favourably to both the most recent shallow embedding methods (QuatE), and deep embedding methods (SACN). Note that with the same decoder (DistMult), MAGNA using its own embeddings achieves drastic improvements over using the corresponding DistMult embeddings.

## 4.3 MAGNA MODEL ANALYSIS

Here we present (1) the spectral analysis results, (2) effect of the hyper-parameters on MAGNA performance, and (3) attention distribution analysis to show the strengths of MAGNA.

**Spectral Analysis: Why MAGNA works for node classification?** We compute the eigenvalues of the graph Laplacian of the attention matrix $\mathbf{A}$, $\hat{\lambda}_i^g$, and compare to that of the diffused matrix $\mathcal{A}$, $\lambda_i^g$. Figure 3 (a) shows the ratio $\hat{\lambda}_i^g/\lambda_i^g$ on the Cora dataset. Low eigenvalues corresponding to large-scale structure in the graph are amplified (up to a factor of 8), while high eigenvalues corresponding to eigenvectors with noisy information are suppressed (Klicpera et al., 2019b).

**MAGNA Model Depth**. Here we conduct experiments by varying the number of GCN, GAT and our MAGNA layers to be 3, 6, 12, 18 and 24 for node classification on Cora. Results in Figure 3 (b) show that both deep GCN and deep GAT (even with residual connection) suffer from degrading performance, due to the over-smoothing problem (Li et al., 2018; Wang et al., 2019a). In contrast, the MAGNA model achieves consistent best results even with 18 layers, making deep MAGNA model robust and expressive. Notice that GAT with 18 layers cannot out-perform MAGNA with 3 layers and $K$=6 hops, although they have the same receptive field.

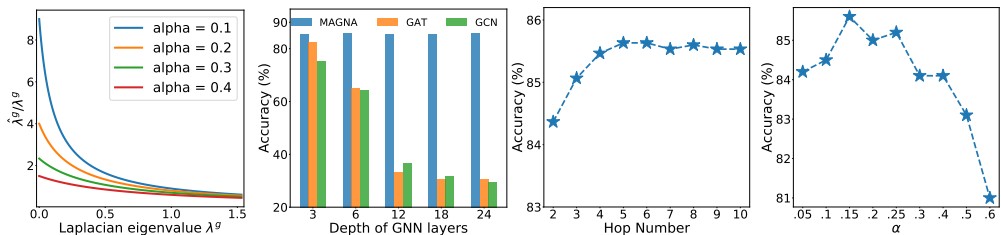

Figure 3: Analysis of MAGNA. (a) Influence of MAGNA on Laplacian eigenvalues. (b) Effect of depth on performance. (c) Effect of hop number $K$ on performance. (d) Effect of teleport probability $\alpha$.

**Effect of $K$ and $\alpha$.** Figures 3 (c) and (d) report the effect of hop number $K$ and teleport probability $\alpha$ on model performance. We observe significant increase in performance when considering multi-hop neighbors information ($K > 1$). However, increasing the hop number $K$ has a diminishing returns, for $K \geq 6$. Moreover, we find that the optimal $K$ is correlated with the largest node average shortest path distance (e.g., 5.27 for Cora). This provides a guideline for choosing the best $K$.

We also observe that the accuracy drops significantly for larger $\alpha > 0.25$. This is because small $\alpha$ increases the low-pass effect (Figure 3 (a)). However, $\alpha$ being too small results in the model only focusing on large-scale graph structure and ignores too much high-frequency information.

**Attention Distribution**. Last we also analyze the learned attention scores of GAT and MAGNA. We first define a discrepancy metric over the attention matrix $\boldsymbol{A}$ for node $v_i$ as $\Delta_i = \frac{\|\boldsymbol{A}_{[i,:]} - U_i\|}{\text{degree}(v_i)}$ (Shanthamallu et al., 2020), where $U_i$ is the uniform distribution score for the node $v_i$. $\Delta_i$ gives a measure of how much the learnt attention deviates from an uninformative uniform distribution. Large $\Delta_i$ indicates more meaningful attention scores. Fig. 4 shows the distribution of the discrepancy metric of the attention matrix of the 1st head w.r.t. the first layer of MAGNA and GAT. Observe that attention scores learned in MAGNA have much larger discrepancy. This shows that MAGNA is more powerful than GAT in distinguishing important and non-important nodes and assigns attention scores accordingly.

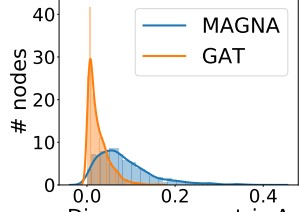

Figure 4: Attention weights on Cora dataset.

## 5 RELATED WORK

Our proposed MAGNA belongs to the family of Graph Neural Network (GNN) models (Battaglia et al., 2018; Wu et al., 2020; Kipf & Welling, 2016; Hamilton et al., 2017), while taking advantage of graph attention and diffusion techniques.

**Graph Attention Neural Networks (GATs)** generalize attention operation to graph data. GATs allow for assigning different importance to nodes of the same neighborhood at the feature aggregation step (Veličković et al., 2018). Based on such framework, different attention-based GNNs have been proposed, including GaAN (Zhang et al., 2018), AGNN (Thekumparampil et al., 2018), GeniePath (Liu et al., 2019b). However, these models only consider direct neighbors for each layer of feature aggregation, and suffer from over-smoothing when they go deep (Wang et al., 2019a).

**Diffusion based Graph Neural Network**. Recently Graph Diffusion Convolution (GDC) (Klicpera et al., 2019b;a) proposes to aggregate information from a larger (multi-hop) neighborhood at each layer, by sparsifying a generalized form of graph diffusion. This idea was also explored in (Liao et al., 2019; Luan et al., 2019; Xhonneux et al., 2019; Klicpera et al., 2019a) for multi-scale deep Graph Convolutional Networks. However, these methods do not incorporate attention mechanisms which proves to have a significant gain in model performance, and do not make use of edge embeddings (e.g., Knowledge graph) (Klicpera et al., 2019b). Our approach defines a novel multi-hop context-dependent self-attention GNN which resolves the over-smoothing issue of GAT architectures (Wang et al., 2019a). EdgeNets (Isufi et al., 2020) also extends attention mechanism for multi-hop information aggregation, but it needs more parameters to compute the attention scores of multi-hop neighbors. In contrast, our method infers the attention scores of multi-hop neighbors based on the one-hop neighbor attention scores via graph diffusion, and thus not only more parameter efficiency but show better spectral property.

## 6 CONCLUSION

We proposed Multi-hop Attention Graph Neural Network (MAGNA), which brings together benefits of graph attention and diffusion techniques in a single layer through attention diffusion, layer normalization and deep aggregation. MAGNA enables context-dependent attention between any pair of nodes in the graph in a single layer, enhances large-scale structural information, and learns more informative attention distribution. MAGNA improves over all state-of-the-art methods on the standard tasks of node classification and knowledge graph completion.

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
