# OpenReview forum: "Multi-hop Attention Graph Neural Network"
_ICLR.cc/2021/Conference — Reject_

### Official Review · AnonReviewer3 · 2020-10-23
**Interesting combination of graph attention and graph diffusion**

**Rating:** 6
**Confidence:** 4

**Review:**

==== Summary ====
This paper proposes MAGNA, a multi-hop self-attention mechanism for attention based graph neural networks. The proposed method increases the receptive field at each layer, requiring less layers to achieve a large receptive field. Also, with the proposed method the attention coefficient between two nodes is not just a function of the two nodes but also of their neighbourhood. The proposed MAGNA method is an extension of GAT networks that introduces a diffusion step on the computed attention coefficients, following a similar approach (Diffusion-GCNs) that has been used for GCNs.

==== Pros: ====
* The proposed method extends GAT layers to have multi-hop receptive fields without increasing the number of model parameters.
* Even though the two main building blocks (Graph Attention and Graph Diffusion) of the paper are not novel, their combination is novel and it achieves state of the art empirical results in two different tasks.
* The evaluation of MAGNA and comparison with previous approaches is well done, using two different tasks and the standard benchmarks for those tasks.
* The paper is well written.


#### Cons:
* The proposed method MAGNA seems to be similar to the APPNP method proposed in [Klicpera 2019a] that also uses diffusion to increase the neighbourhood around each node in a GCN layer (without attention). However the two models are not compared and the later is not included in the Related Work section, even though it is cited previously.
* The ablation study is a good idea, but it isn’t very clear how it is done.
* The motivation behind including layer normalization and deep aggregation or why they are useful isn’t entirely clear.

#### Questions:
1) MAGNA has 3 main differences compared to GAT: layer normalization, diffusion and deep aggregation. In the ablation study in Table 1, which components are removed for each row? When it says “No LayerNorm”, does that mean that it uses diffusion and deep aggregation but no LayerNorm? If so, is there an ablation test where MAGNA just uses the diffusion step but without the other two components? That would be very relevant, since it would be like a GAT network with a larger receptive field at each layer, and would show how much improvement the increased receptive field brings without the other components.
2) Since the paper puts a lot of emphasis in the multi-hop capability of MAGNA, and related to my previous question, it would be quite interesting to compare MAGNA against a GAT network that has a larger receptive field, for example by using a multi-hop adjacency matrix computed with the diffusion process from [Klicpera 2019b] (called sparsified matrix in that paper) instead of the 1-hop adjacency matrix used in the original GAT paper. With this comparison we could see the benefit of using the diffusion of the attention values instead of just increasing the receptive field of GAT by allowing each node to pay attention to nodes up to K hopes away.
3) In section 4.1 the authors say that “Since MAGNA computes many attention values, layer normalization is crucial in ensuring training stability”. Doesn’t MAGNA compute the same number of attention values as GAT and it then diffuses them? Any ideas why layer normalization is crucial when diffusion is used but not without it? (as seen in the ablation study, when MAGNA doesn’t use diffusion the scores are on par with GAT, which suggests that layer normalization is only useful with diffusion).

#### Minor comments
In the Edge Attention Computation paragraph in section 2.2, the authors say “To compute representation of” and probably meant “To compute the representation of”.

$W_o$ in Equation 6 is not explained in the text.

Across the paper and in the caption for Figure 2, the text talks about MAGNA blocks, but the text in the Figure says “DAGN” Block. Is that the same? Same in the 2nd plot in Figure 3, it says DAGN, should it be MAGNA?

#### Reasons for score
The authors give a solid justification for using a diffusion process to increase the receptive field of GNNs with attention, and along with previous papers on graph diffusion provide a good motivation and explanation of their method. Also, the experiments section shows that MAGNA achieves state of the art results in two different tasks. Because of these reasons, I vote for accept. My only concern is that some components don’t have a clear justification besides the empirical improvement in performance (layer normalization and deep aggregation) and that comparisons with other techniques (APPNP and GAT with k-hop adjacency matrix) would make the experiments section stronger.

----- Post Rebuttal Update -----
After the author's rebuttal and the discussion with other reviewers, I have decided to lower my score to 6.
The reason is that during the review and discussion process it was pointed out that the main contribution of the paper is more incremental than novel, as both multi-hop GATs and diffusion for GNNs have been explored before in similar ways. Additionally, even though the authors provide some theoretical grounding for their work, some of it is a bit disconnected from the rest of the paper, like the relation with PageRank introduced in section 3.2, which is not referenced or discussed in any other section.

---

> ### Author Response · Authors · 2020-11-22
> **Response to Reviewer #3**
>
> We thank the reviewer for insightful feedback and for noting that our​ method “give a solid justification for using a diffusion process to increase the receptive field of GNNs with attention”. We added new ablation studies as suggested by the reviewer and clarified the motivation of some key modules below.
>
> Q1: APPNP method proposed in [Klicpera 2019a] that also uses diffusion… … not compared and the later is not included in the Related Work section, even though it is cited previously.
>
> R1: We thank the reviewer for pointing out this related work. We added the comparison of our method to APPNP in Table 1 and also cited it in our related work in the revised version. MAGNA still achieves significant improvement (average of 1.8%) over APPNP.
>
> Q2: The ablation study is a good idea, but it isn’t very clear how it is done… is there an ablation test where MAGNA just uses the diffusion step but without the other two components?
>
> R2: We explored the ablation study on diffusion, layer normalization and feed-forward layer by removing one component each time, which indicates the most gain comes from the diffusion. According to the suggestion, we have added the ablation study by removing both ‘LayerNorm’ and ‘feedforward’ layers (Please refer to ‘No (LayerNorm + Feed-Forward)’ line in Table 1). The result further confirms the effectiveness of our proposed multi-hop attention method.
>
> Q3: The motivation behind including layer normalization and deep aggregation or why they are useful isn’t entirely clear.
>
> R3: The motivation of these two components is as follows. For ‘‘deep aggregation”, we can view it as a learnable activation function compared to elu activation in GAT. The reason is that ‘two layer MLP’ can approximate many different functions (please refer: Pinkus, Allan. "Approximation theory of the MLP model in neural networks." Acta numerica 8.1 (1999): 143-195.) Therefore, compared to the ‘shallow’ elu in GAT, we can view MLP as a learnable deep aggregation method.
>
> For “Layer normalization”, we make use of the attention matrix in a recursive way (multiple multiplication) to calculate the diffused attention (please see Eq. 5). Layer normalization is helpful to stabilize the learning procedure as it is very effective at stabilizing the hidden state dynamics in recurrent computation procedure (please refer: Ba, J. L., Kiros, J. R., & Hinton, G. E. (2016). Layer normalization. arXiv preprint arXiv:1607.06450.) We have included the demonstration in the paper.
>
> Q4: Compare MAGNA against a GAT network that has a larger receptive field.
>
> R4: We show that MAGNA with just 3 layers and 6 hop wide attention per layer significantly  outperforms GAT with 18 layers, even though both architectures have the same recep-tive field. Please refer to the depth analysis in Fig. 3(b).
>
> Q5: Doesn’t MAGNA compute the same number of attention values as GAT and it then diffuses them? Any ideas why layer normalization is crucial when diffusion is used but not without it?
>
> R5: We thank the reviewer for the question. The reviewer is right, we compute the same number of attention values in GAT. We make use of the attention matrix in a recursive way (multiple multiplication) to complete the attention diffusion (Please refer to Eq. (5) in section 2.2, page 4). Layer normalization is very effective at stabilizing the hidden state dynamics in recurrent networks (see Ba, J. L., Kiros, J. R., & Hinton, G. E. (2016). Layer normalization. arXiv preprint arXiv:1607.06450.).
>
> Q6: “To compute representation of” and probably meant “To compute the representation of”.
>
> R6: We thank the reviewer for carefully checking on our paper, we have revised that sentence.
>
> Q7: ‘Wo’ in Equation 6 is not explained in the text.
>
> R7: We thank the reviewer for checking our paper carefully, we have explained that in the revised version.
>
> Q8: Name mis-match in Name mis-match in Figures 2 and 3… it says DAGN, should it be MAGNA?
>
> R8: We thank the reviewer for checking our paper carefully, we have revised the notation in these figures and fixed such a mis-match.

---

### Official Review · AnonReviewer4 · 2020-10-26
**Official Blind Review #4**

**Rating:** 6
**Confidence:** 3

**Review:**

The authors propose a novel attention-based GNN called MAGNA. The main contribution consists in considerably increasing the receptive field by considering a multi-hop neighborhood instead of the standard one hop. The technical challenge consists in obtaining attention scores for all relevant nodes in an efficient way. MAGNA solves this by using a diffusion-based technique combined with a geometric distribution. The authors show that the latter further allows for approximations, and also give interesting theoretical insights (e.g., show a relation to page rank).

The paper is overall well written, related work is considered adequately, the idea is interesting, and the results are convincing. The evaluation comprises several different datasets, domains, tasks, and competitive baselines. The ablation studies give interesting insights in the effects of different parameter choices. Altogether, I suggest to accept the paper.

----------------------------------------------
Smaller comments:
- p.3: "degenerate categorical distribution with 1 category": I think this should be explained.
- p.3: "effectively creating attention shortcuts between nodes that are not connected (Figure 1).": I assume you mean "directly connected" since the whole approach seems to consider only connected nodes?
- p.4: "as well as good model generalization.": How does the diffusion process ensure good model generalization?
- p.5: Footnote 2: ??
- 4.2. Baselines: The paragraph mentions few what is not in the table, so maybe you can just drop it and use the space for more descriptions.
- 4.2. Results.: The last sentence is unclear to me.

----------------------------------------------
Update after Rebuttal: I have read the other reviews and authors' responses.

While I do think the novelty of the contribution is sufficient, given that  the paper referenced in another review has not been peer-reviewed yet, the new ablation results in Table 1 show that the paper's contribution is not outstanding. I adjusted my score.

---

> ### Author Response · Authors · 2020-11-22
> **Response to Reviewer # 4**
>
> We thank the reviewer for the positive and detailed feedback. We are glad that the reviewer likes our approach. Below, we address the reviewer’s concern on some concepts and results.
>
> Q1: Clarification of ‘degenerate categorical distribution with 1 category’
>
> R1: If there is no relation type over the edge, we can view that there is only one ‘pseudo’ relation type (and thus N_r = 1). In fact, for edge attention computation in Eq. (1), we omit the component of W_r*r_k if there is no relation type in the graph. We have clarified this in the revision.
>
> Q2: Results.: The last sentence is unclear to me. MAGNA using its own embeddings achieves drastic improvements over using the corresponding DistMult embeddings.
>
> R2: We make use of MAGNA as an encoder to learn the entity embedding. That is, the node embedding of the last layer of MAGNA over the knowledge graph. Based on the learned entity embedding, we score each triple (h, r, t) in the knowledge graph by the score function in the well-known DistMult KG embedding model. For standard DistMult, the entity embedding is randomly initialized and the fine tuned with DistMult score function, which requires the scores of positive triples should be greater than those of negative triples.
>
> Q3: How does the diffusion process ensure good model generalization
>
> R3: First, the diffusion process has the same parameter efficiency in attention computation compared to GAT; moreover, the spectral analysis of the diffusion process shows that such diffusion can be viewed as a lower-pass filter (focus on large graph structure and more robust to noisy graph).
>
> Q4: Footnote 2: missing reference
>
> R4: Thanks for checking our paper carefully, we have fixed it in the revised version.

---

### Official Review · AnonReviewer2 · 2020-10-26
**Novelty in the Spectral Analysis**

**Rating:** 5
**Confidence:** 5

**Review:**

Summary:

This paper mainly proposes to learn attention-based edge coefficients by incorporating information from farther away nodes by means of their shortest path (powers of the adjacency matrix). Furthermore, the authors show the spectral properties of the proposed algorithm and its equivalence to personalized page rank.

Strong points:

The numerical experiments show slight improvement.

The spectral analysis is potentially interesting.

Weak points:

The multi-hop attention network has been done before (see below). The novelty thus resides only in the spectral analysis and the page rank equivalence.

Recommendation:

The paper is ok, but I do not consider it to have enough novelty to be considered for publication in ICLR.

Major comment:

1) Assuming there is only a scalar assigned to each edge (i.e. \mathcal{R} = \mathbb{R}), then (1),(3),(4) are a particular case of (39)-(41) in Isufi et al, 2020, where A_{k} = alpha I for k<K and A_{K} = H (the H matrix in eq. 4). This renders (1), (3), (4)'s only novelty to be the potential for vector-valued edge weights. Please elaborate on the comparison with Isufi et al, 2020 in the paper.

E. Isufi, F. Gama, and A. Ribeiro, "EdgeNets: Edge Varying Graph Neural Networks," arXiv:2001.07620v2 [cs.LG], 12 March 2020. [Online]. Available: http://arxiv.org/abs/2001.07620

2) Why is R in N_r x d_r? Shouldn't it have N_e as a dimension? So if we have 5 edge types (like in the QM9 molecule data set, with 4 bond types and an extra no-bond type, encoded as one hot vectors), that means that R has only 5 rows? And what does the number of columns represent? It's just that X is very straightforward: number of nodes x number of channels. But R is hard to interpret.

3) The LHS of eq. (2) shows dependence with i, j and l. The RHS shows dependence with i, j, l, and also k. Where does k appear in the matrix indexing? Is there a different attention score matrix for each value of k?

4) I believe the paper would benefit for an increased elaboration on the usefulness of proposition 2, since this is the main novelty. Let us say we are given a graph, and we choose to describe that graph by means of some matrix (either the adjacency or the Laplacian). The choice of graph description fixes the frequency interpretation of the graph. Different matrix choices lead to different frequency interpretations. Now, the attention mechanism changes this matrix description by learning a new matrix description. However, this new, learned, matrix description will most likely not even share the same eigenvectors as the original matrix description of the graph. So how do we know that the filters learned are actually low-pass filters in the graph? They maybe low-pass filters in the learned graph by attention mechanisms, but they may not be in the actual graph that was given by the problem. Please, clarify what's the interpretation of Proposition 2, since matrices A and \mathcal{A} are being learned from data.

Minor comments:

Due to Cayley-Hamilton theorem, there is no need for (3) to go to infinity. It suffices for it to go to N_n-1.

Reference to Sandryhaila and Moura, 2013 is missing at the beginning of Section 3 when referring to discrete signal processing on graphs.

A. Sandryhaila and J. M. F. Moura, "Discrete signal processing on graphs," IEEE Trans. Signal Process., vol. 61, no. 7, pp. 1644–1656, 1 Apr. 2013.

Footnote 2 on page 5 is missing a proposition reference.

--- UPDATED SCORE ---

First of all, I would like to thank the authors for carefully addressing my comments.

In light of the authors' response, and after a thorough and careful discussion with the other reviewers, I have decided to update my score to 5 (five).

In summary, I appreciate the authors' effort to signal the differences between their work and Elvin et al. While I agree with these, I still think this is only an incremental contribution. For further reference, after carefully discussing the paper with other reviewers, these two published papers were also pointed out where multi-hop attention is addressed. Namely,
https://openreview.net/forum?id=rkKvBAiiz
https://ieeexplore.ieee.org/document/8683050
I apologize for not finding these papers in my first round of reviews, but it does not really alter my evaluation of the paper.

I think that the most novel contribution is on the spectral analysis. But this is only stated, with no real insights developed, and not emphasized enough. More insight on this would definitely bring a novelty. More specifically, novelties that would have made the paper more interesting: (i) a different way of computing the attention coefficients, that would be more parameter efficient (as opposed to eq. (1)) in the paper, (ii) actual useful insights into what the frequency response of the learned filters look like in the attention matrix as opposed to the given support matrix of the graph, (iii) a comparison between the spectral basis of the learned attention matrix as compared to the support matrix.

---

> ### Author Response · Authors · 2020-11-22
> **Response to Reviewer #2**
>
> We thank the reviewer for acknowledging the novelty of our spectral analysis. We also thank the reviewer for pointing out the related work, E. Isufi, F. Gama, and A. Ribeiro, "EdgeNets: Edge Varying Graph Neural Networks," arXiv:2001.07620v2 [cs.LG], 12 March 2020. [Online]. Available: http://arxiv.org/abs/2001.07620, w.r.t. multi-hop attention. Below we clarify the difference of our method and EdgeNets and also add the discussion to our paper. We also clarify some unclear points proposed by the reviewer and fix them in our revision.
>
>
> Q1: Comparison with Isufi et al, 2020 of “EdgeNets’’ in the paper.
>
> R1: We thank the reviewer for pointing out this related paper of “EdgeNets’’. The Eqs. 41, 42 in this paper also make use of multi-hop information for node feature learning via attention mechanism. However, EdgeNet is different from our MAGNA as following:
>
> First, the attention computation over the powers of the matrix is different. In “EdgeNets”, they compute the attention scores over A, A^2, A^3, ... respectively (according to Eqs, 41, 42) based on node features with different parameters (see the e_k in Eq. 41).
>
> In comparison, our MAGNA first computes the one-hop attention matrix A based on node features, and then infers the attention scores of A^2 and A^3, ... based on A. Therefore, (1) our method is more parameter efficient as there is no need for more parameters for attention computation over A^2, A^3, …; And (2) our multi-hop attention matrix computed by Eq. (3) has the same eigein-vectors as those of one-hop attention matrix A: this is crucial to ensure the effect of lower-pass filtering compared to one-hop attention matrix A. In summary, our proposed MAGNA not only shows better spectral property but also more parameter efficiency, compared to EdgeNets.
>
> Moreover, except for the attention mechanism, we design a more effective GNN model structure (attention diffusion + layernorm + feedforward), and also supply an efficient implementation and theoretical analysis.
>
> Due to the lack of open-source code, we have not run the evaluation of EdgeNets in the standard benchmark. However, note that we have already compared to 20 baselines in this paper, including many multi-hop architectures (Diffusion-GCN, JK-Net, APPNP).
>
> We thank again for the reviewer pointing out this related paper and we have added it in our related work.
>
> Q2: Clarification of R in N_r x d_r and k in Eq. (2)
>
> R2: N_r means the number of edge/relation types in the graph. R represents the embeddings of N_r ‘edge/relation’ types, and d_r demonstes the dimension of each relation type embedding.
> In Eq (2), k means the index of relation between node i and node j, if we compute attention over graph with different edge type/edge attribute (such as for KG, there are different relations over the edges), r_k means the k-th relation type, and \mathbold{r}_k means the embedding of k-th relation type, and k <= N_r.
>
> We have clarified this in the revision.
>
>
> Q3: How do we know that the filters learned are actually low-pass filters in the graph?
>
> R3: Here we would like to clarify a misunderstanding: The conclusion of the lower-pass filter is based on the comparison of multi-hop diffusion matrix to learned one-hop attention matrix (i.e., the attention matrix in GAT), rather than to the original adjacency matrix. Comparison to attention matrix is more relevant here since we already know that attention improves graph learning in many tasks.
> We infer the diffused multi-hop attention matrix based on the one-hop attention matrix, that is, the multi-hop attention matrix is a linear combination of different powers of one-hop attention matrix and the linear coefficient is defined as theta in Eq. (3). This leads to the low-pass conclusion in the paper. We have clarified this in the revision.
>
> Q4: Missing reference to Sandryhaila and Moura, 2013 at the beginning of Section 3 when referring to discrete signal processing on graphs.
>
> R4: Thanks for checking our paper carefully and pointing out such a missing reference, we have added this reference in the revised version.
>
> Q5: Footnote 2 on page 5 is missing a proposition reference.
>
> R5: Thanks for checking our paper carefully, we have fixed this in the revised version.

---

### Official Review · AnonReviewer1 · 2020-10-29
**Important problem but the proposed solution may be not powerful than existing models**

**Rating:** 5
**Confidence:** 5

**Review:**

Summary:

     Conventional Graph Neural Networks (GNNs) learn node representations that encode information from multiple hops away by iteratively aggregating information through their immediate neighbors. Self-Attention modules have been adopted to GNNs to selectively aggregate information coming through the immediate neighbors at different propagation stages. However, current self-attention mechanisms are limited to only attend over the nodes' immediate neighbors and not directly over their neighbors that are multiple hops away. Here in this work, the authors intend to address this issue and propose a means to obtain attention scores over indirectly connected neighbors.

    The message passing paradigm is commonly adopted in GNNs because directly computing the higher powers of an adjacency matrix is not scalable. The same scalability concern is present for this work, which tries to obtain attention scores for indirect neighbors directly. Thus in order to solve this issue, the authors propose to diffuse the learned attention scores from their 1-hop neighbors to neighbors that are multiple hops away, thereby providing a means to directly obtain attention scores over indirect neighbors that are reachable from the nodes.

——
Pros:

	The paper is well written.
	The paper provides experimental results for both homogenous and multi-relational graphs.

——
Concerns:
	(i) Proposed methodology being more powerful than GAT is arguable:

	When the attention scores for indirectly connected neighbors are still computed based on the immediate neighbors' attention scores, it is not convincing enough to be argued as more powerful than GAT, which learns attention scores over contextualized immediate neighbors.  Also, the approximate realization of the model described in Eqn: 5 follows a message-passing style to propagate attention scores. Suppose it is to be argued that standard message-passing-based diffusion is not powerful enough to get a good immediate neighbor representation that encodes neighbors' information from far away. In that case, it is not immediately clear how a similar diffusion, when used for propagating attention scores from immediate neighbors to neighbors multiple hops away, will be more powerful.

(ii) Experimental results are not conclusive:

		(a) Effect of Layer-Norm and FeedForward
			One of the important ablation model that is missing is MAGNA without feed-forward and layer-Norm components. Currently, it is not clear how much of an improvement is achieved because of these standard two components.
		(b) Disentangling the effect of page-rank from the attention diffusion
			Since Diffusion-GCN is also based on Page-Rank based propagation, it would be helpful to compare with the Diffusion-GCN model with these two components appended to them, along with residual connection if already not present. This would help us clarify how much of the gain in performance depends on the page-rank-based propagation compared to the attention propagation. The teleport probability of Diffusion-GCN should also be similarly experimented with and the analysis should be compared with the plots in Figure 3.

 		(c) Comparable or not significant gains achieved in Node classification tasks.
			It is amendable that the authors have reported results for both single-relational and multi-relational graphs. However, the node classification results are not significantly better than GAT or Diffusion GCN on the reported smaller datasets (Table 1) with a single train/val/test split. And on OGB Arxiv dataset, GAT and Diffusion GCN numbers are not reported. Hence, it would be helpful to analyze additional datasets.
  					Ignoring the benefits of LayerNorm that the MAGNA can leverage, comparing its No-Feed-Fwd version with Diffusion-GCN, which is also based on a page-rank formulation, MAGNA gain ~1% improvement on Cora and Pubmed dataset whereas it falls behind by ~1% in Citeseer.

		(d) Disentangling the effect of Multi-scale diffusion from attention diffusion
			Since MAGNA uses a multi-scale diffusion at each layer, a comparison with a similar non-attentive multi-scale diffusion model like Lanczosnet that is also referred in the paper would be helpful to disentangle	and understand the importance of the attention mechanism.

		(e) KG Completion: Missing baselines and model variations

 		- Missing comparison with Self-attention (GAT) based knowledge graph embedding model, KBGAT.
		  Nathani et al., Learning Attention-based Embeddings for Relation Prediction in Knowledge Graphs, ACL 2019
		- Additionally, it would be helpful to have similar model ablation studies of MAGNA model as in Table 1 for KG Completion.

		(f) Depth Analysis:
			  Diffusion-GCN comparison missing. The performance stabilization over GAT might be arising because of the restart probability. MAGNA only has weights associated with 3 layers, unlike with GAT, which has weights for every propagation step.


——
Questions during rebuttal:

	- Kindly clarify concern (i)
	- Check experimental concerns above for additional ablation and baseline variants that is required to disentangle and appreciate the usefulness of the primary contribution, the attention diffusion component.
	- Comparison with the KB-GAT model that is based on GAT for KG completion task, will strengthen the results on KG completion task.
	- Comparison with GAT and Diffusion-GCN with LayerNorm and FeedFwd components on multiple train/test/val splits for smaller datasets or for other datasets from OGB will strengthen the results for node classification.

--- Post-rebuttal
I thank the authors for responding to all the questions and getting back with additional experiment results.

Major concern: While I understand the motivation and how having attention scores over nodes multiple hops away can be powerful, I'm still not convinced with the approximate realization. It is not clear how diffusing attention defined over 1-hop neighbors is powerful over attention methods defined over immediate neighbors that contain k-hop information aggregated from diffusion.

Also, the performance drop and overfitting issue with GAT or diffusion-GCN can be combated similarly by sharing weights across GNN layers and also using a higher-order diffusion matrix at each GNN layer.

---

> ### Author Response · Authors · 2020-11-23
> **Response to Reviewer # 1**
>
> We thank the reviewer for thorough and constructive comments. Based on the reviewer's valuable feedback, we clarified our method and added additional experiments, which further validate the efficacy of our method. We believe these results further strengthen our work.
>
> Q1: It is not immediately clear how a similar diffusion, when used for propagating attention scores from immediate neighbors to neighbors multiple hops away, will be more powerful.
>
> R1: We thank the reviewer for this important question. The reviewer points out that GAT can be stacked many layers to achieve attention towards far-away nodes. There are two reasons why the multi-layer GAT is not as powerful as MAGNA. First, stacking GAT causes oversmoothing as illustrated in the introduction of our paper. Second, GAT uses learnable weight matrices in each layer and capturing larger neighborhoods requires an even larger number of learnable parameters, which empirically results in overfitting. As Fig. 3 illustrates, GATs perform better when they are shallow, in which case attention to far-away nodes cannot be captured.
>
> In contrast, in a single layer, MAGNA collects multi-hop information for feature aggregation. The diffusion process serves as a prior for multi-hop attention weights, and resolves the above challenges. Fig. 3 shows that MAGNA improves performance with increasing layers, significantly out-performing GAT.
>
> Q2: it is not clear how much of an improvement is achieved because of LayerNorm and feedforward
>
> R2: We want to point out the misunderstanding about the ablation table: we observe that without diffusion (the no-diffusion line in Table 1), the performance is very similar to GAT with LayerNorm/feedforward, indicating that diffusion is the crucial factor, and the rest are minor factors. However, for clarity, we have added additional experiments of ablation study by removing layerNorm and feedforward (Pls refer to next-to-last line in Table 1).
>
> Q3: how much of the gain depends on the page-rank-based propagation (diffusion-GCN) compared to the proposed attention propagation
>
> R3: We thank the reviewer and conducted additional experiments by adding both layerNorm and feedforward to Diffusion GCN to make a fair comparison between page-rank-based propagation and attention propagation. The accuracy of Diffusion GCN (PPR) with layerNorm and Feedforward over Cora, Citeseer and Pubmed is 83.4, 72.3 and 78.1, respectively, which is lower than MAGNA’s accuracy over Cora (85.4), Citeseer (73.7) and Pubmed (81.4).  Please refer to Table 6 in the revised Appendix.
>
> Q4: Comparison on multiple train/test/val splits for smaller datasets for node classification tasks
>
> R4: We want to point out that random splitting is often problematic in graph learning (see paper 'Open Graph Benchmark: Datasets for Machine Learning on Graphs') as they are impractical in real scenarios, and can lead to large performance variation. Here we use the standard split, as performed in most of the important related literature including GCN, GraphSAGE, GAT, ClusterGCN, LGCN, ARMA, SGCN, etc. We further record standard deviation to demonstrate statistical significance of experimental results. For completeness, we added additional experiments to compare baselines with our method over random split in the Appendix. We observed that the standard deviation of the accuracy from random split is much larger than that from standard split (2.4 v.s. 0.2 in PubMed). Thus standard split is preferred by the community. Please refer to Table 7 in the Appendix for details. Moreover, we have also added two multi-hop GNNs baselines: JKNet and  GaAN (attention guided), over large OGB arxiv data in Table 2, where JKNet and GaAN’s accuracies are 72.19 (std=0.21) and 71.97(std=0.24), and our MAGNA’s accuracy is 72.76 (std = 0.14).
>
> Q5: Comparison with the KB-GAT model that is based on GAT for KG completion task
>
> R5: We thank the reviewer for pointing out. We intentionally did not compare with KB-GAT because in the widely accepted paper on RotatE, it indicates a bug in ConvKB’s evaluation (which KB-GAT is based on). Please refer to https://openreview.net/forum?id=HkgEQnRqYQ about “Not mention results of ConvKB…” part of RotatE OpenReview discussion (ICLR 2019).
>
> Moreover, we used the latest implementation by the first author of the ConvKB paper (please refer: https://github.com/daiquocnguyen/ConvKB). The MRR numbers over FB15k-237 reported in Github is 0.302, a significant drop from 0.396 in the original KB-GAT paper. In contrast, MAGNA’s MRR is 0.369, which is much higher than 0.302.
>
> Q6: Depth Analysis for Diffusion-GCN
>
> R6: We conducted additional experiments of depth analysis of Diffusion-GCN compared to MAGNA, we observed that deep Diffusion GCN (with residual connection) suffers from degrading performance, and unlike MAGNA, the classification of Diffusion-GCN with 18 layers is 34.6 which is a significant drop from two layer Diffusion-GCN (83.6). Please refer to Fig. 1 in appendix in the revision.

---

### Decision · Program_Chairs · 2021-01-07
**Final Decision**

**Decision:**

Reject

**Comment:**


This paper has been reviewed by four knowledgeable referees. Two of them slightly leaned towards acceptance, whereas the other two suggested rejection. The main issues raised by the reviewers were (1) limited novelty [R1,R2], (2) missing baselines and ablations [R1,R3], (3) limited insights on the spectral analysis [R2], and (4) missing motivation behind modeling choices [R1,R3]. The rebuttal included a number of experiments requested by the reviewers (e.g. ablation with diffusion only [R1,R3], extended Diffusion GCN [R1], APPNP baseline [R3]), and adequately motivated some of the modeling choices.

The central question of the reviewers' discussion was whether the contribution of this paper was significant enough or too incremental. The discussion emphasized relevant literature which already considers multi-hop attention (e.g. https://openreview.net/forum?id=rkKvBAiiz [Cucurull et al.], https://ieeexplore.ieee.org/document/8683050 [Feng et al.], https://arxiv.org/abs/2001.07620 [Isufi et al.]), and which should have served as baseline. In particular, the experiment suggested by R3 was in line with some of these previous works, which consider "a multi-hop adjacency matrix " as a way to increase the GAT's receptive field. This was as opposed to preserving the 1-hop adjacency matrix used in the original GAT and stacking multiple layers to enlarge the receptive field, which as noted by the authors, may result in over-smoothed node features. The reviewers acknowledged that there is indeed as slight difference between the formulation proposed in the paper and the one in e.g. [Cucurull et al.]. The difference consists in calculating attention and then computing the powers with a decay factor vs. increasing the receptive field first by using powers of the adjacency matrix and then computing attention. Still, the multi-hop GAT baseline of [Cucurull et al.] could be extended to use a multi-hop adjacency matrix computed with the diffusion process from [Klicpera 2019], as suggested by R3. In light of these works and the above-mentioned missing baselines, the reviewers agreed that the contribution may be viewed as rather incremental (combining multi-hop graph attention with graph diffusion). The discussion also highlighted the potential of the presented spectral analysis, which could be strengthened by developing new insights in order to become a stronger contribution (see R2's suggestions).

To sum up, this was a very discussed paper, where the reviewers ultimately reached a consensus to reject, with no strong opposition. I agree with the reviewers' assessment and therefore must reject. I encourage the authors to follow the reviewers' suggestions and consider the multi-hop baselines as well as the hints provided by the reviewers about the spectral analysis to strengthen their work.